# Trends in Admissions and Outcomes at a British Wildlife Rehabilitation Centre over a Ten-Year Period (2012–2022)

**DOI:** 10.3390/ani14010086

**Published:** 2023-12-26

**Authors:** Elizabeth Mullineaux, Chris Pawson

**Affiliations:** 1Capital Veterinary Services Ltd., Haddington, East Lothian EH41 4JN, UK; 2Secret World Wildlife Rescue, Highbridge, Somerset TA9 3PZ, UK; 3Department of Animal and Agriculture, Hartpury University, Hartpury, Gloucestershire GL19 3BE, UK; chris.pawson@hartpury.ac.uk; 4College of Health, Science and Society, University of the West of England, Bristol BS16 1QY, UK

**Keywords:** hedgehog, rescue, rehabilitation, triage, wildlife, welfare

## Abstract

**Simple Summary:**

Millions of animals pass through wildlife rehabilitation centres globally each year. Wildlife centre databases can provide an evidence base for treatment and contribute to conservation. Records of British animals admitted to a centre over a 10-year period were analysed. Birds were more frequently admitted than mammals, reptiles, and amphibians, and nine species predominated the admissions; hedgehogs were the most common species admitted. Most admissions were in the summer and spring months, and juvenile animals were admitted more frequently than ‘orphans’ or adults. ‘Orphaned’ was also the predominant reason given for admission, followed by ‘injured’. A total of 42.6% of animals were eventually released back into the wild, 19.2% died in captivity, and 37.2% were euthanised. The outcome was better for orphaned animals than those admitted because of injury. Unexpected natural deaths in captivity were found to decline over the period of study, consistent with improved early triage. These findings can be used to focus training and seasonal resources on the species and case types most likely to be successful. The findings also have the potential to contribute to our understanding of anthropogenic impacts, historical and regional variations in ecosystem health, and resultant implications for animal welfare.

**Abstract:**

Millions of animals pass through wildlife rehabilitation centres (WRCs) globally each year, some dying in captivity, others euthanised, and some released into the wild. Those caring for these animals are generally well-intentioned, but skills, knowledge, and resources may be limited, potentially compromising animal welfare. WRC databases provide an opportunity to provide an evidence base for treatment and conservation efforts. 42,841 records of animals admitted over a 10-year period to a British WRC were analysed. More birds (69.16%) were admitted than mammals (30.48%) and reptiles and amphibians (0.36%). Most admissions were in the summer (48.8%) and spring (26.0%) months. A total of 9 of the 196 species seen made up 57% of admissions, and hedgehogs were the most common species admitted (14% of all admissions and 20% of mammals). Juvenile animals (35.5%) were admitted more frequently than ‘orphans’ (26.0%) or adults (26.4%). ‘Orphaned’ was also the predominant reason for admission (28.3%), followed by ‘injured’ (25.5%). 42.6% of animals were eventually released back to the wild, 19.2% died in captivity, and 37.2% were euthanised; 1% of outcomes were unknown. The prognosis was better for orphaned animals than for those admitted because of injury. Unexpected natural deaths in captivity were found to decline over the period of study, consistent with improved early triage. These findings can be used to focus veterinary and WRC training and seasonal resources on the species and case types most likely to be successfully rehabilitated and released. The findings also have the potential to contribute to our understanding of anthropogenic impacts, historical and regional variations in ecosystem health, and resultant implications for animal welfare.

## 1. Introduction

Wildlife rehabilitation is defined as the treatment and temporary care of injured, diseased, and displaced indigenous animals and the subsequent release of healthy animals to appropriate habitats in the wild [1]. It is estimated that millions of animals are rescued and rehabilitated globally each year [2,3] by individuals and wildlife rehabilitation centres (WRCs). Wildlife rehabilitators have hugely variable skills and knowledge, and WRCs vary enormously in their facilities [4,5]. Reasons for undertaking wildlife rehabilitation are usually welfare-driven but may also include considerations such as providing a public service, interacting with nature, and personal fulfilment [3,4,5,6,7,8,9]. Furthermore, wildlife rehabilitation is also recognised as having a public educational role regarding specific anthropogenic threats to wildlife [10,11,12], as well as a role in monitoring ecosystem health [13,14,15] and the impacts of urbanisation [16,17]. Although wildlife rehabilitation has generally been considered to only have indirect impacts on conservation rather than population-level impacts [18], there is now increasing recognition of the potential for the discipline to have direct impacts on conservation [19,20,21].

Wildlife rehabilitation practices are not entirely without potential deleterious consequences. Negative impacts on individual animal welfare are recognised [4], including stress associated with captivity [3,22,23,24]. There is also a potential for disease transmission within wildlife centres [25] and to wildlife [26] and domestic animals [27] upon release, dissemination of antimicrobial resistance [28], and human–wildlife pathogen transmission [26,29]. Negative psychological pressures have been noted for both wildlife rehabilitators [5,6] and veterinary professionals [30,31], whose concerns and goals relating to wildlife care may differ [6]. Recent developments in the regulation of wildlife rehabilitation in countries including Australia [2,4,7,32], Canada [6], and the USA [33] may help to address some of these issues. However, even where regulation is in place, frustrations still exist around variations in standards, training, and the availability of funding [5,6,7,34,35,36,37]. Many of the disagreements around wildlife rehabilitation arise from a lack of evidence regarding best treatment practices and outcomes for casualty animals and captive-reared juveniles. In recent years, studies from Australia [2,16,38], South Africa [39], the USA [33,40], and Chile [36] have considered outcomes for animals admitted to WRCs in those countries. A useful international review of the available data has also been published [3]. 

### The United Kingdom Context

There are over 600 ‘wildlife rescues’ listed in the United Kingdom (UK) [41]. These figures may, however, be a gross underestimation because wildlife rehabilitation is currently unregulated, and there are no official figures. In 2011, it was estimated that at least 71,000 wild animals were admitted to wildlife establishments in England and Wales [42], a figure more than twice that estimated in an earlier publication [43]. As acknowledged in a review of the available literature [44], there is a paucity of data relating to wildlife centre admissions and releases in the UK, with the most frequently cited figures dating back to annual returns to the British Wildlife Rehabilitation Council (BWRC) in the late 1990s [45,46]. An analysis of data from four WRCs run by the Royal Society for the Prevention of Cruelty to Animals (RSPCA) between 2000 and 2004 attempted to predict release rates for eight species covering a range of taxonomic groups that were commonly admitted for rehabilitation [43]. Other publications are limited to single taxa or post-release monitoring [42]. There is no state funding for wildlife rehabilitation in the UK, financial support is limited, and data collection can be poor, especially outside of the larger charities. To the best of the authors’ knowledge, this is the first study to present such a large data set from the UK and allows comparison with similar international studies.

Complete and good-quality data on wildlife admission and outcome data are essential in assisting wildlife rehabilitators to plan and provide appropriate resources, especially where funding is limited. Knowing which casualties (species, reasons) are admitted when (time of year) and which cases are successful (likelihood of release) can inform approaches to triage and focus resources and staff training towards those animals with the best possible outcomes. Treatment methods and assessments for release (physical health and behaviour) can then be critically reviewed. This information can also be used to target education appropriately for others involved, such as veterinary professionals (vets and vet nurses/techs), government staff, and conservationists. Contributions to conservation, both direct and indirect, can then be made. For example, known anthropogenic reasons for admission could impact decisions by local authorities and the central government. There is also a key function of identifying trends in the proportion of different animals admitted and contributing to other data sets that can contribute to our longer-term understanding of changes in species and ecosystem health. From an animal welfare perspective, unrealistic expectations of outcomes for wildlife casualties (and, therefore, potentially unnecessary suffering) are compounded by the lack of scientific evidence. Baseline information on WRC admissions and returns to the wild is key to understanding not only which cases are likely to be successful but also those that should be euthanised at the first opportunity. With that in mind, the current study reports on admissions and outcomes at a British WRC over a ten-year period from 2012 to 2022.

## 2. Materials and Methods

Secret World Wildlife Rescue (SWWR) is a large WRC in Somerset in the southwest of England (located at 51°12′21.4′′ N and 2°57′49.6′′ W). The WRC is in a rural area, close to the Somerset levels, but has a catchment area that includes the cities of Bath and Bristol. The WRC can offer emergency care to all species of wild birds, terrestrial mammals, reptiles, and amphibians. Animals are brought directly to the WRC by members of the public or are collected by volunteer drivers following an initial telephone conversation. Some animals are referred by local veterinary practices, and a small number are from other WRCs. Animals are assessed upon admission by wildlife care staff, trained in the primary examination and triage of British wildlife, and receive appropriate veterinary intervention. Animals that are unsuitable for eventual release back to the wild are euthanised at the first appropriate opportunity. Those considered to be suitable for treatment and rehabilitation, where there is a strong likelihood that with appropriate care, they will eventually be clinically and behaviourally fit to be released back to the wild, are kept at the WRC until released.

In the summer of 2011, SWWR developed a Microsoft Access database v 14 to record individual animal admission and outcome data (Appendix A). Trained staff have access to the database, and each animal is assigned an individual log number. ‘Class’ of animal is selected from a three-option dropdown menu with ‘reptiles and amphibians’ grouped together. ‘Species’ of animal is selected at either genus or species level as identifiable. There is a category for the sex of the animal where this can be determined. An ‘age’ category is allocated to each animal based on its stage of development: ‘orphan’, where animals are neonates or dependent juveniles; ‘juvenile’, when the animal is immature but independent; and ‘adult’, where animals are mature. ‘Reason for admission’ is selected from seven possible options using a ‘best fit’ approach. ‘Natural causes’ include weather-related reasons or naturally occurring diseases. ‘Other’ includes a known reason for admission that does not fit into one of the six defined options (e.g., entrapped but apparently uninjured); as with other categories, ‘unknown’ can also be used where there is no known reason for the admission. ‘Final outcomes’ are ‘euthanased’, natural death in captivity (‘died’), or ‘released’ back to the wild. No animals are kept in long-term captivity; animals with no likelihood of eventual release back to the wild are euthanised. Only those animals with an outcome at the end of 2021 were included in the data analysis. ‘Time to end result’ is calculated in days from the date of admission to the date of the outcome.

Over the decade of the study period, there were inevitably some external factors that potentially influenced admission; these included the COVID-19 pandemic of 2020/21 and the start of the H5N1 avian influenza epidemic affecting the UK in the winter of 2021. These impacts were not quantified but are discussed. The significant organisational change was staffing stability, improved training, and experience, which appeared to improve the ‘triage’ of animals and confidence in early euthanasia. These factors could not be quantified but are considered in terms of their impact on outcomes. 

The database was considered over a 10-year period from 1 January 2012 to 31 December 2021. The Microsoft Access database was transferred to Microsoft Excel with categories as shown in Appendix A, which were used for further analysis. The records were manually checked, any domestic animals (including ‘exotic pets’) were removed, any obvious errors (e.g., wrong ‘class’ of animal for species recorded) were corrected, and an Excel formula was used to calculate the number of days in captivity based on the date of admission and the date of the final outcome. The data were then imported into SPSS v28 (IBM Corporation, Armonk, NY, USA) for analysis.

## 3. Results

### 3.1. Total and Class of Animals Admitted

Over the 10-year study period, 42,841 animals were admitted: 29,629 (69.16%) birds, 13,057 (30.48%) mammals, 88 reptiles (0.20%), and 67 amphibians (0.16%). As the numbers of reptiles and amphibians admitted were very small, these classes were grouped together (Figure 1 and Appendix A). 

### 3.2. Changes in Total Admissions over the Study Period

The total number of annual admissions rose gradually from 3806 individual admissions in the first year of the study in 2012 to a ten-year high of 5170 in 2018. Over the ten-year study period, the mean number of admissions per year was 4282 (SD = 685.22) (Figure 2 and Appendix A). Birds were consistently the most frequently admitted class, ranging from a low of 64% of all admissions in 2012 to a high of 73% of all admissions in 2018. Annual admissions decreased in 2019 (by 17%) and 2020 (by 28%), the latter of which was partly due to the COVID-19 pandemic and associated ‘lockdown’ restrictions in the UK. It is important to note that there were some other noteworthy year-on-year fluctuations during the study period, particularly related to the proportion of birds vs. mammals admitted each year. These are presented in Table 1 and Table 2 and discussed below.

### 3.3. Changes in the Class of Animal over the Study Period

The increase in the total number of animals admitted each year in the first four years of the study was largely driven by a 42% increase in the admissions of birds (rising from 2452 in 2012 to 3474 in 2016) (Table 1). During the same time period, mammal admissions also rose, but by a more modest 9% (from 1335 in 2012 to 1455 in 2016). There was then a period of relative stability in the number of admissions between 2015 and 2017. However, in 2018, annual admissions rose again, and once again, this was attributable to a rise in avian admissions (+12%), while mammal admissions declined (−7%). Z tests revealed that this relative increase from 2017 to 2018 in the proportion of birds admitted (70% of all admissions rising to 73%) was a significant proportional change (z = 4.07, *p* < 0.0001). 

The ten-year high in admissions in 2018 was followed by a significant decline in 2019, and this was again found to be driven by changes in avian admissions. Between 2018 and 2019, avian admissions declined by −23%, compared with a 4% increase in admissions of mammals (Table 2). Z tests revealed that this relative decrease in the proportion of bird admissions from 2018 to 2019 (73% falling to 66% of all admissions) was significant (z = 7.29, *p* < 0.0001). Finally, admissions of both birds and mammals fell between 2019 and 2020 (by −28% and −27%, respectively), before both increasing by 13% between 2020 and 2021. This very similar decline and then increase across both birds and mammals is likely to be linked to the COVID-19 pandemic and associated lockdowns in the UK in 2020 and 2021. The avian flu epidemic in the winter of 2021 may also have impacted the number of species admitted at this time, although the full impact of this was not quantified.

### 3.4. Seasonal Variation in Admissions

Admissions were grouped to reflect the UK seasons of autumn (Sept/Oct/Nov), winter (Dec/Jan/Feb), spring (Mar/Apr/May), and summer (Jun/Jul/Aug) (Figure 3 and Appendix A). 20,916 (48.8%) of admissions were in the summer, which saw 53.7% of all bird admissions, 37.8% of mammals, and 36.8% of reptiles/amphibians admitted in that season. A total of 11,141 (26.0%) admissions were in the spring, and 7027 (18.3%) were admitted in the autumn. Winter saw the lowest admissions, with only 2957 (6.9% of all admissions). Amphibian and reptile numbers never exceeded 1% of the total number of animals admitted in the season and were therefore excluded from inferential analysis. 

A 4 (season) × 2 (class) Chi-Square analysis revealed a significant association between season and class (χ^2^(4) = 1640, *p* < 0.001). While bird and mammal admissions are relatively balanced in autumn (52% and 48%, respectively), they then diverge significantly, with birds increasing their proportion of total admissions from 52% in autumn to 65% of all admissions in winter and increasing to 70% in spring, and finally accounting for 76% of all admissions in summer (Table 3). Z tests revealed that the increased proportion of avian admissions from season to season was significant for all seasonal changes. 

### 3.5. Species Admitted

Admissions included 196 identified species: 144 species of birds, 43 mammals, 6 reptiles, and 3 amphibians (Appendix A). Some animals were only identified by class or genus. A small number of domestic animals, including ‘exotic pet’ species, were admitted that were assumed to be living wild, recently escaped, or abandoned. Nine species had more than 1000 individuals admitted over the study period. These species totalled 24,599 animals, or 57% of all admissions. Hedgehogs were by far the most common species admitted, making up 14% of admissions and 20% of mammals. UK conservation status [47,48] was also noted (Table 4).

### 3.6. Sex of Animal

The sex of the animal was very rarely recorded, and, therefore, these findings were not analysed.

### 3.7. Age of Animal Admitted

When the admissions were classified based on age, 11,141 (26.0%) were identified as ‘orphans’, 15,234 (35.5%) as ‘juveniles’, and 11,298 (26.4%) as ‘adults’. Age was ‘unknown’ in 5168 (12.1%) of cases. Age at admission was also considered according to the class of animal (bird, mammal, and reptile/amphibian) (Figure 4 and Appendix A). Relative to birds and mammals, reptiles and amphibians were less likely to be admitted as orphans or juveniles and more likely to be adults or unknown.

### 3.8. Reason for Admission

Reasons for admission included 10,909 (25.5%) animals that were ‘injured’, 89 (0.2%) were ‘poisoned/polluted’, 12,124 (28.3%) were ‘orphaned’, 1131 (2.6%) due to ‘natural causes’, 9772 (22.8%) for ‘other’ reasons, 4159 (9.7%) had been ‘caught by cat’ and 429 (1.0%) ‘caught by dog’. For 4228 (9.9%) cases, the reason for admission was ‘unknown’. The reason for admission was also considered according to the class of animal (Table 5).

Presumed direct anthropogenic causes (injured, poisoned/polluted, caught by cat, and caught by dog) together made up 36.4% of all known reasons for admission—41.3%, 25.2%, and 86.0% of admissions of birds, mammals, and reptiles/amphibians, respectively.

### 3.9. Reason for Admission by Season

Reasons for admission were considered for each season, with the four most commonly known reasons illustrated in Figure 5 and Appendix A. Admissions for all reasons were highest in the summer, followed by the spring. A total of 55.8% of orphaned animals were admitted in the summer, and 31.2% were admitted in the spring; 45% of injured animals and 50% of animals caught by cats were admitted in the summer, and 23.7% and 33.5%, respectively, were admitted in the spring.

### 3.10. Overall and Animal Class Outcomes

When outcomes for the animals admitted were considered, 15,932 (37.2%) were euthanised, 8234 (19.2%) died naturally in captivity, and 18,232 (42.6%) were released back into the wild. For 443 (1.0%) cases, the outcome was unknown; this included any animals transferred to other centres. Domestic, hybrid, and non-indigenous animals that could not be released under Section 14 of the Wildlife and Countryside Act 1981 (including some included in the Invasive Alien Species (Enforcement and Permitting) Order 2019) either died, were euthanised, or were appropriately transferred elsewhere. The outcome was also considered according to the class of animal (Figure 6 and Appendix A). A Chi-Square analysis revealed that birds were more likely to be euthanised and less likely to be released relative to mammals (χ^2^(1) = 713.45, *p* < 0.001).

### 3.11. Reason for Admission vs. Outcome

The outcome was considered for the top three known reasons for admission in all classes: birds and mammals (Figure 7 and Appendix A). Injured animals were more likely to be euthanised (62.5%) than released (23.8%); 64.0%, 57.2%, and 52.2% of injured birds, mammals, and reptiles/amphibians, respectively, were euthanised. Orphaned animals were more likely to be released (59.9%) than euthanised (15.9%); 57.2%, 62.1%, and 66.7% of orphaned birds, mammals, and reptiles/ amphibians, respectively, were released.

### 3.12. Outcomes over Time

The final outcome for animals admitted to the WRC was considered over the study years (Appendix A). The proportion of total animals that were released remained stable over the study period. There was, however, an evident trend for an increase in the proportion of animals euthanised, while a decreasing proportion died in captivity (Figure 8). Between 2012 and 2021, the incidence of euthanasia increased from 22.9% to 46.8%, while natural death in captivity decreased from 29.9% to 11.1%. A Chi-Square analysis revealed that the increase in euthanasia rate and decrease in the rate of natural deaths in captivity between 2012 and 2021 were significant (χ^2^(1) = 669.20, *p* < 0.001).

### 3.13. Time in Captivity

The time in captivity ranged from 0 to 535 days for mammals, 0 to 441 days for birds, and 0 to 261 days for reptiles/amphibians. The figures relating to the duration of time in captivity were not normally distributed (KS > 0.05), as a result of those animals who had a final outcome achieved on day 0. The median number of days in captivity was 3.00 overall: 2.00 for birds, 6.00 for mammals, and 0 for reptiles/amphibians. For animals that were euthanised, the mean number of days in captivity was 3.08 (±11.64) overall (median = 0); 2.56 (±9.73) for birds; 4.89 (±16.49) for mammals; and 0.79 (±2.80) for reptiles/amphibians. For animals that naturally died in captivity, the mean number of days to that outcome was 4.93 (±12.37) overall (median = 1); 4.16 (±10.22) for birds; 6.44 (±15.66) for mammals; and 1.50 (2.36) for reptiles/amphibians. For animals that were eventually released, the mean number of days to that outcome was 36.09 (±37.30) overall (median = 27); 29.55 (±26.14) for birds; 48.01 (±49.23) for mammals; and 8.19 (32.30) for reptiles/amphibians. A total of 13,764 (32.46%) animals had a final outcome on day 0; 10,464 (76.0%) were euthanised; 2190 (15.9%) died; 1102 (8.0%) were released; and 8 (0.1%) had an unknown outcome. If animals with a day 0 outcome are excluded, the mean number of days in captivity was 26.12 (±34.35) overall (median 15.00). 

## 4. Discussion

The findings of this study represent those of just one UK WRC; however, they are the first known large report of data from the UK and, as such, allow useful comparison with other international studies. Although all centres will have some species biases as a result of both geography and centre special interests (for example, badgers at SWWR [27]), the data are considered to be broadly representative of most UK centres. It is hoped that this data set will, however, allow for a more detailed comparison with other UK data in the future. 

### 4.1. Total and Class of Animals Admitted

The number of animals admitted to a WRC facility will vary enormously depending on geography and capacity. The findings from SWWR are typical of a larger centre in the UK. An annual average of 2636 animals were admitted to 27 centres in 2011 [43], although admissions of some species have increased subsequently [49]. Some UK centres report dealing with nearly 20,000 cases a year [50] with up to 1600 animals on a site at any one time [51], but it is unclear to what extent these cases are made up of telephone advice only, animal admissions, and long-term resident animals. In this study, no animals originating in the wild were kept in long-term captivity. The total number of animals admitted to WRCs across the world is frequently described as ‘millions’ [2,3], but there is no accurate supporting data. The Australian Zoo and Wildlife Hospital (AZWH) is one of the largest WRC facilities in the world and cares for up to 8000 wildlife admissions annually [16]. Confiscated animals, resulting from illegal possession or transport of protected species, significantly impact the numbers and species admitted to some WRCs [36,52,53,54,55]. Such incidents are rare in the UK, although some centres keep long-term captive animals, which is controversial [32,42,43]. The large numbers of animals admitted to WRCs mean that adequate provision of staff, facilities, and funding is required to ensure adequate care and animal welfare. It is important, as centres grow and intake increases, that funding and resources increase proportionally in order to avoid any negative welfare impacts. 

When divided by taxonomic class, the findings of this study were very similar to 16,000 UK WRC admissions in the late 1990s, which consisted of 67·1% birds, 32·5% mammals, and small numbers of reptiles and amphibians [45,46]. Admissions to WRC according to class of animal vary internationally and within countries, reflecting differences in topography. Avian species, which are typically more numerous, usually predominate admissions. As many admissions are because of direct anthropogenic impacts (see 4.8), the extent of overlap of animal populations with human activities will also impact admission. In Australia, birds accounted for 53.4% of admissions in New South Wales [2] and 51.1% of admissions in Queensland [16]. In Tasmania, however, mammals predominated (58.5%), with just 38.2% birds [38]. Birds dominated admissions to WRCs in Catalonia, Spain (89%; [52]); Athens, Greece (83.3%; [12]); Chile (86.0%; [36]); and South Africa (90%; [39]). Conversely, in the USA, admissions to WRC in New York State were only slightly higher in birds (51.9%) than mammals (43.7%) [33], and in two centres in Oregon, birds comprised just 28.0% and 37.1% of admissions [11]. The number of reptiles and amphibians admitted to WRCs is usually very low: 4.3% in New York State [33], 0.3% in Catalonia [52], 1.7% in Chile [36], and 2% in South Africa [39]. Only in Australia are the cited reptile admissions higher, with figures of 19.5% of admissions in New South Wales [2] and 14.4% of admissions in Queensland [16] being recorded. These varied findings show that the skills of wildlife rehabilitators and veterinary professionals internationally will need to focus on slightly differing areas of expertise in order to best support wildlife rehabilitation in the countries involved.

### 4.2. Changes in Admissions over the Study Period

Changes in admissions to WRCs may be impacted by environmental disasters such as oil spills and wildfires [3,16,56,57]. No such events occurred in the UK during the present study period. The COVID-19 pandemic of 2020/2021 reduced staffing capacity at SWWR as well as changing public interactions with UK wildlife. In 2020, people were initially ‘locked down’ and unable to go outside of their homes, which would have reduced both the occurrence of negative human actions (road traffic collisions, etc.) as well as the ability of people to find and report injured wildlife. This was, however, followed by a time when people were off work and encouraged to exercise in public outdoor spaces and the countryside. These changes had a variety of impacts on wildlife, with increased human disturbance in some environments, including parks and gardens, alongside increased opportunities for feeding around picnic sites [58]. The overall human–animal interaction increased. Public appreciation and concern for wildlife also increased during this time [59] and may have contributed to the upward trend in admissions into 2021. The only significant outbreak of wildlife disease in the UK was the H5N1 avian influenza epidemic, which began in the winter of 2021 [60] and resulted in some birds not being admitted because of associated species-specific risks; these were not quantified. 

### 4.3. Changes in Class of Animal over the Study Period

The changes in avian admissions in 2013 and 2019 cannot be accounted for in operational terms. The authors, therefore, recommend that future comparative analyses explore temporally similar changes in other regions, including potential weather/climate and as yet unrecognised disease factors. Further species-specific analyses would also be of interest.

### 4.4. Seasonal Variation in Admissions

In this study, most admissions of all species were in the UK summer and spring, an effect seen across all categories of admission and not restricted to ‘orphans’. Such trends may be a function of seasonal population expansion and high-risk behaviours and activities related to breeding. Much lower admissions were seen in the autumn and winter months. This may be because of reduced animal activity, but also a possible reduction in human–animal interactions that lead to animal admissions in the colder autumn and winter months. This is consistent with the findings from other studies showing admissions to WRCs peak in the spring and summer, with lower numbers in autumn and winter [2,12,16,38,39,40,52]. In the northern hemisphere, admissions are typically highest in April and May and lowest in December [40], whilst in the southern hemisphere, admissions peak in October and November and are lowest from May to July [2,39]. 

Seasonal trends in admissions have clear implications for planning and resourcing services in WRCs and veterinary practices [44]. Staff will be required in the winter months for maintenance as well as to care for those animals admitted, but the numbers required will be lower than in the spring and summer. The use of seasonal staff is an obvious solution, but staff training and experience will need to be ensured.

### 4.5. Species Admitted

Although many different identified species were admitted to SWWR during the study period, admissions of the nine most common species accounted for over half of all admissions. Both birds and mammals that were commonly seen are ‘red’ and ‘amber listed’ species of conservation concern [47,48]. This is of significance as although these species are considered ‘common’ in a wildlife rehabilitation context, their numbers are dwindling, and WRCs have the potential to contribute both to direct species conservation through treatment, rehabilitation, and release, as well as to surveillance of both disease and anthropogenic potential causes of species decline. Previous information from WRCs across the UK found four common species accounted for over 40% of admissions: hedgehogs, feral pigeons, blackbirds, and collared doves [45,46]. These species were also common in the current study, with wood pigeons and herring gulls being new additions to the list. In the current study, hedgehogs were the most commonly admitted mammals, a similar finding to the previous study, where this species accounted for 16% of all admissions and 54% of mammals [45,46]. Geographical differences between SWWR and other UK WRCs may be an explanation for subtle differences in findings, together with changes over time; further data analysis is required to assess these potentially interesting effects.

These findings illustrate that although the range of wildlife species admitted to a WRC is large, some species are commonly seen, with several of these being conservation concerns. Veterinary and wildlife rehabilitation professionals can focus on facilities and equipment, as well as staff training on these species, in particular in the UK on hedgehogs and common species of birds. 

### 4.6. Sex of Animal Admitted

Although the study database has an option for recording the sex of the animal, this was infrequently recorded and not included in the data analysis. There were several reasons for this: often, the database was completed before the animal was fully examined, reception staff were not trained in the sex determination of all species, some animals could not be safely examined to determine their sex without chemical restraint, and some animals (especially birds) were not sexually dimorphic. This is consistent with studies in other WRCs where sex is either not included in data analysis [2,16] or is frequently not determined [36,52]. 

### 4.7. Age of Animal Admitted

In the current study, over half of all animals and all mammalian and avian admissions were immature. Relative to birds and mammals, reptiles and amphibians were less likely to be admitted as orphans or juveniles and more likely to be adults or unknown. Around a third of birds and mammals were admitted solely for the reason of being ‘orphaned’. In previous studies in the UK, 50% of bird and 54% of mammal admissions were of immature animals; of these, 32% of birds and 27% of mammals had no injuries, and the primary reason for their admission to the WRC was that they were ‘orphans’ [46]. The slightly higher proportion of mammals admitted in the current study may be because of the geographical location of SWWR or a reputational bias towards the hand rearing of orphan mammals. Differences over time, including increased anthropogenic pressures, may also play a part. In other UK surveys, 65% of polecat (*Mustela putorius*) admissions [61] and 68% of wood pigeon admissions [62] were juveniles. 

The term ‘orphan’ is used extensively in the literature; however, the limitations of this term with respect to the true cause of a dependent animal being admitted to a WRC have been acknowledged [42,52], and similar limitations are likely in the current study, where the three options for selection of age (orphan, juvenile, or adult) were open to some interpretation by staff. Admissions of immature animals are common internationally. In mainland Spain (Catalonia), 54% of WRC admissions were first calendar-year animals, with ‘orphaned’ being the second most prevalent category of admission [52]. In Gran Canaria, the orphaned young made up 27.19% of avian admissions [52]. In Athens, Greece, ‘orphans’ were also the second most common category, making up 24.8% of admissions [12]. Most European hedgehogs admitted to WRCs in the Czech Republic were ‘hoglets’ (59.5%) [63], and most common kestrels were ‘nestlings’ (34.7%) [64]. In North America, ‘orphans’ accounted for 37.3% of admissions in New York State (Hanson et al., 2021) and 21.6% in Oregon [11]. In a WRC in Virginia, USA, 23% of grey foxes (*Urocyon cinereoargenteus*) and 33% of red foxes (*Vulpes vulpes*) were admitted as orphans [65]. In Canada, orphaned or abandoned young accounted for 25% of bird and 66% of mammal admissions [6]. In Australia, orphans accounted for 24.6% of admissions in Queensland [16], 20.1% in New South Wales [2], and 16.5% in Tasmania [38]. In a WRC in South Africa, most of the animals admitted (43%) were juveniles, contributing to 48%, 30%, and 36% of admitted birds, mammals, and reptiles, respectively. The reason for admission for 17% of birds and 11% of mammals was ‘young’ [39].

The large numbers of young animals admitted are likely to reflect abandonment due to human disturbance, injury/death of adults rearing young, or, to some extent, ‘natural wastage’ in r-strategist species. Many of the young animals admitted are uninjured but require both facilities and experienced staff to rear them appropriately to an age at which they could be released. These skills are usually those of wildlife rehabilitators rather than veterinary professionals, although the latter should be able to provide emergency care and then quickly move the animals to an appropriate WRC. Importantly, young animals need to be appropriately released back to the wild, often ‘soft released’, which again requires the time, skills, and specialist knowledge of rehabilitation professionals [66]. Without such provisions, animal welfare may be compromised.

### 4.8. Reasons for Admission

In this study, reasons for admission were classified into categories based on a ‘best fit’ approach following a conversation between the finder and a staff member. A similar approach based on primary admitting/presenting signs has been used by other authors [43]. Although animal ecology, biology, and behaviour appear to predispose certain species of animals to certain threats [16], and one might expect some variation in admissions according to the geographical location of the WRC, studies from WRCs across the world showed some common trends. Although there are some differences in the groupings of reasons for admission, ‘orphans’ are numerous in most studies and were the second most common cause for admissions in many studies [2,12,16,33,67,68]. The primary reason for admission was frequently some classification of ‘injury ‘or ‘trauma’ or ‘anthropogenic causes’; for example, 34.6% of admissions in Athens, Greece, were classified as an ‘unknown accident’ [12]. 

#### 4.8.1. Reasons for Admissions: ‘Injury’

In this study, ‘injured’ was the second most common reason for admission after ‘orphaned’. Traumatic injury is a common finding in animals presented to WRCs internationally, with ‘trauma’ accounting for 38.1% of WRC admissions in New York State [33]. A recent study of data both from Canadian WRCs and submissions to a national pathology service found the main reasons for admissions in both instances, 44% and 48% of admissions, respectively, to be trauma [69], illustrating the importance of trauma on both WRC admissions and truly wild populations. In this study, as in others, trauma in birds is frequently reported to be a more significant cause of morbidity and mortality than in mammals. This is assumed to be because of the causes and types of injuries sustained by birds, but it requires further analysis. Trauma was involved in 81.1% of the birds admitted to a WRC in Florida, USA [70]. Previous studies in the UK found traumatic injuries to account for 39% of all admissions, 43% of bird casualties, and 30% of mammal casualties [45,46]. Trauma was the most frequently observed cause of admission in all animals (35.8% in birds, 23.2% in mammals, and 27.8% in reptiles) in Chile [36]. Trauma was the third most important reason for admission to a WRC in Catalonia, Spain, making up 17.4% of admissions overall, but it accounted for 71% of admissions in waders, 60% in birds of prey, 59% in herons and allies, and 41% in carnivores [52]. Trauma also accounted for 27.8% of morbidity in all birds in Gran Canaria, Spain [53] and 18.14% of morbidity, specifically in seabirds [71]. Several studies considering raptor admissions alone, in the UK [17,72] and elsewhere [64,67,68,73,74,75,76,77], found trauma to be one of the two most common reasons, alongside orphans, for admissions. Trauma is also commonly cited as a reason for admissions to WRCs for specific species of mammals. In a study in Virginia, USA, trauma was the cause of morbidity and mortality in 46% of grey foxes and 27% of red foxes [65]. In the UK, 40% of adult hedgehogs were admitted with ‘trauma’ [49]. 

In the current study, the range of injuries sustained by animals admitted to SWWR included trauma from vehicle collisions, window collisions, falls from a height, and trauma from a garden or farming equipment, although this detail was recorded in clinical records rather than the database. Other authors have classified admission reasons relating to injury in different or more detailed ways. Several authors cite ‘collision with vehicles’ (‘hit by car’, ‘road traffic collision’, ‘vehicle impact’, etc.) as a common anthropogenic cause of trauma, accounting for between 11% of admissions across all species in a Canadian study [69], 34.7% in an Australian study [16], and 30% in Tasmania [38]. Of the known causes for rescue in New South Wales, Australia, ‘collisions with vehicles’ were the most common across all taxonomic groups, accounting for 24.3% of all cases, 20.5% for birds, and 33.5% for mammals [2]. Window or building strikes are also commonly cited as traumatic reasons for admissions and account for 5% and 7% of admissions to a pathology service and WRCs, respectively, in Canada [69]. Collisions with buildings or windows accounted for 29.7% of the known reasons for the admission of avian casualties in Florida [70]. 

This study and others internationally suggest that trauma is a very common reason for wildlife casualty admissions. These findings are important for training wildlife professionals to deal with commonly occurring reasons for the presentation of animals, which should focus on the skills to examine and triage casualty animals efficiently to decide if treatment is both possible and in the animal’s best interest. The skills required for treatment, if appropriate, will then be good veterinary trauma management skills, which, in many instances, can easily be extrapolated from domestic animals. Types, locations, and levels of occurrence of traumatic incidences can also inform conservation efforts [20,21] and should be one reason for collaboration between WRCs and conservationists. 

#### 4.8.2. Reasons for Admission: Poisoned/Polluted

The number of admissions in the current study in the ‘poisoned/polluted’ grouping was very small. These findings are similar to those from other WRCs, where clinical poisoning is infrequently reported. Although species such as raptors are regarded as being good sentinels of exposure to and effects of chemicals in the environment [15,78], such effects are often chronic rather than presenting with acute clinical signs [79]. In one study in Tenerife, only 2.4% of a study of raptors were poisoned [75]. Hedgehogs have also been shown to be frequently exposed to environmental toxins [14], yet a study of hedgehogs in Portugal found only 1.6% were considered to have been poisoned [80]. In some species, for example, seabirds, poisoning/intoxication has been reported as a significant (24.69%) reason for admissions [71]. The true impact of poisons and pollutants on wildlife, such as oil [81], is unknown but does not appear to be a common direct reason for admissions to most WRCs. Chronic low-grade exposure to toxins may, however, predispose to other causes of morbidity and mortality [15] and, therefore, WRC admissions. Further pathological investigations are required to fully assess the significance of toxicities and inform both WRC care and conservation efforts.

#### 4.8.3. Reasons for Admissions—Dog or Cat Attack

Capture by cats or dogs accounted for significant admissions of animals across all classes, with cat attacks predominating. This is consistent with other studies suggesting interactions with domestic pets are significant for conservation [82]. Estimates suggest that free-ranging domestic cats kill 1.3–4.0 billion birds and 6.3–22.3 billion mammals annually in the USA [83], with concerns of a similar impact in Australia [16]. In the UK, an estimate of 229 million prey animals per annum brought home by domestic cats has been made [84]. An earlier questionnaire of cat owners found 71 different species of animals being caught by their pets, of which 69% were mammals, 24% were birds, and 5% were reptiles and amphibians [85]. Although it has been suggested that many injured animals are released by cat owners [86], many are brought to WRCs. Previous studies in the UK found admissions due to cat trauma to account for 13% of bird admissions to WRCs, 5% of mammals, and 24% of reptiles and amphibians [46]. A cat attack was found to have occurred in 21% of adult wood pigeons and 16% of juveniles [62]. 

Similar trends are seen internationally; in Italy, 14.2% of admissions to a WRC were because of injury from human impacts, with 54.3% of these being due to predation by mainly cats [87]. A study of WRCs in North America found domestic pets responsible for 14% of admissions and the second most common identifiable cause of wildlife injury [10]. Interactions with cats represented 12.3% of admissions to two WRCs in Oregon, USA, and were again the second most common reason for admissions [11]. Attacks by cats accounted for 25.4% of known causes of trauma in birds admitted to a WRC in Florida [70]. In a WRC in Tennessee, 20% of cases were due to interactions with domestic pets, 14% were cat-related, and 6% were with dogs. In a WRC in Madison, Wisconsin, dog and cat interactions accounted for 9.7% and 5.6%, respectively, of small mammal and bird admissions, with birds admitted more commonly because of cat interactions and mammals because of dog interactions [88]. In Canada, cat attacks accounted for 23% and 13% of bird and mammal admissions, respectively, in one study [6] and, more recently, 6% of overall admissions to WRCs [69]. In South Africa, dog/cat attacks accounted for 13% of admissions in both mammals and birds [39]. In all studies with available data, mammals and birds are admitted throughout the year because of cat and dog interactions. More cat and dog attacks were seen at a WRC in Wisconsin in the breeding seasons in the spring and summer, and fledgling birds were also being admitted more frequently than adults and hatchling birds [88], suggesting vulnerability when first leaving the nest. Another study in the USA in Virginia, however, found that although cat interactions were most frequent in juvenile mammals (40.5%), compared to neonates (34%) and adults (25.5%), in birds, adults with cat injuries were more frequently admitted to the WRC (42.7%) than juveniles (37.2%) or nestlings (20.1%) [89]. Although birds frequently feature in reports of cat and dog attacks, the significance of domestic pet injuries to reptiles and amphibians has been recognised [16,90]. Like birds, bats are also susceptible to cat trauma, especially when leaving roosts in the spring. Cat-related trauma accounted for the main reason for the admission of 28·7% of bat casualties in a study in Italy [91] and around half of the traumatic deaths of bats found in bats in a study in Germany [92]. 

The significance of the high prevalence of domestic pet injuries on native wildlife in the UK and worldwide is highly concerning for both conservation and welfare reasons [84,85]. Cats, which may be domestic pets or feral, are of particular concern [93]. From a veterinary perspective, the treatment of bite wounds in wildlife poses specific problems relating to the injuries caused, difficulties in the treatment of the infections caused by deep puncture wounds, and often a poor prognosis despite treatment [86,94]. Veterinary and WRC staff should be trained in good examination, triage, and appropriate treatment of such injuries.

#### 4.8.4. Reasons for Admissions—Anthropogenic

Combined direct anthropogenic causes made up a large proportion of known reasons for admission in this UK study. These figures are likely to be an underestimation, as factors such as human disturbance of neonates and juveniles are not included. In a systematic review of the literature relating to WRCs, anthropogenic reasons accounted for 48% of studies reporting admissions into care, with the most commonly reported reasons for admissions being collision with motor vehicles, gunshot or poaching, domestic or feral animal attack or predation, oil spill, toxicosis or poisoning, electrocution/collision with powerlines, collision with structures, confiscation, relocation or displacement, and entanglement [83]. Anthropogenic interferences were involved in 64% of admissions in Spain [52]. In the USA, anthropogenic causes in East Tennessee found 31.1% of reasons for admission to be cat-related, dog-related, hit by automobile, and other human encounters leading to trauma [95]. These figures illustrate the significance of direct human activities on WRC admissions, some of which may be highlighted and mitigated through public education. 

#### 4.8.5. Reasons for Admission: ‘Orphaned’

The admission of immature animals to WRCs has been discussed above.

#### 4.8.6. Reasons for Admission: ‘Natural Causes’ 

Naturally occurring causes for admissions, including conditions such as malnourishment and disease, accounted for only a small number of cases in this current study. This finding is consistent with the low numbers of admissions for these reasons in other WRC studies. Only 9.7% of admissions to the AZWH in Queensland had overt disease [16], and only 5.1% in New South Wales [2]. Malnourishment and illnesses accounted for 12.75% and 6.2%, respectively, of avian admissions to a WRC in Florida, USA; the two categories were not considered to be mutually exclusive [70]. In raptors in Spain, less than 10% of admissions were for reasons other than ‘trauma’ or ‘orphan’ [67]. In common kestrels in the Czech Republic, only 4.96% were reported as exhausted/starved [64]. Starvation was reported in 5.6% of nocturnal and 12.1% of diurnal birds of prey in central Italy [68]. In foxes, species where infectious conditions might be expected to be more commonly reported, mange was only seen in 17% of red foxes, 6% had toxoplasmosis, 3% had presumed canine distemper, and 3% of rabies was reported in grey foxes [65]. A Canadian study showed that whilst significant mortalities in submissions to a wildlife pathology service were caused by infections (27%) and emaciation (23%), the same trends were not encountered in admissions to WRCs [69]. This suggests reasons for admissions to WRCs do not necessarily reflect the true reasons for morbidity and mortality in wildlife and instead reflect human–animal conflict and other contact. An inability to diagnose ‘natural causes’ at admission may also result in many of these cases falling into ‘other’ or ‘unknown’ categories. A clinical diagnosis, including a post-mortem examination where appropriate, is needed to differentiate between these admission categories and should be encouraged in all WRCs.

The findings of this and other studies all concur that primary disease is not a major reason for admissions to WRCs. This means that a good knowledge of wildlife diseases is not necessarily required for veterinary professionals to triage, make prognostic decisions, and treat wildlife casualties. This is especially significant in countries such as the UK, where all veterinary professionals are required to provide first aid and emergency care to these animals and may be concerned about their lack of relevant knowledge. 

#### 4.8.7. Reason for Admission: ‘Unknown’

In this study, ‘other’ and ‘unknown’ categories could also be used as reasons for admissions. There are clearly limitations to this methodology, with many animals with ‘natural causes’ potentially being identified in this way. The findings, however, are similar to those in other WRC admission studies and illustrate the limitations of classification at admission. WRCs are often very busy, and history from the finder of the animal may be limited, and assessment frequently involves lay rather than veterinary professional staff [3]. Equally, unlike domestic animals, wildlife casualties come with a very limited clinical history, so some reasons for admission are genuinely unknown. Reasons for admissions to WRCs in mainland Australia [2], Tasmania [38], and South Africa [39] included 54%, 44%, and 31%, respectively, recorded as unknown or undetermined. In a study of raptors admitted to UK rescue centres, 46% of admissions were for ‘unknown’ reasons [17]. To better classify the unknown cases, an improved clinical diagnosis and post-mortem examination would be required. The costs associated with these can, however, be limiting, and further investigations do not always alter the prognosis and individual animal outcomes and are, therefore, often hard to justify for self-funding WRCs.

### 4.9. Reasons for Admissions by Season

The general trend of admissions being highest in the summer and spring was seen across the reasons for admission, with orphans and injured animals predominating. Immature animals in the spring and early summer account for a seasonal trend in other studies [2,39,52], although other seasonal effects in admissions are seen, such as road traffic accidents [2] and reptile admissions [16]. Other species-specific ecology and biology may also result in seasonal variations; for example, admission of debilitated ‘autumn juvenile’ hedgehogs [49,63,96], admissions of badgers related to reproductive activity in that species [27], increased foraging activity, and associated trauma in adult sparrowhawks feeding young [72]. Hunting, including the use of glue traps, was noted as a seasonal reason (both inside and outside closed seasons) for admissions in some studies in mainland Europe [67,68,75,97]. Although persecution of birds, in particular birds of prey, occurs in the UK [98], organised hunting is usually limited to gamebirds and waterfowl during distinct shooting seasons. 

Further species-specific analysis of the data would be extremely useful to investigate these complex seasonal trends, which may have conservation impacts. 

### 4.10. Overall and Animal Class Outcomes

The overall number of animals released back into the wild in this study is not dissimilar to previous smaller studies in the UK that showed overall release rates of 39% [43] and 42% [46]. When attempting to compare the findings of this study with others internationally, several variations in data collection and analysis are evident and potentially confounding. These include the inclusion in other studies of not-caught animals, already dead animals, animals transferred to other rehabilitation facilities, unreleased long-term captive animals, and the different grouping of animals for ‘positive’ and ‘negative’ outcomes [2,3,16,33]. Most admissions to WRCs are anthropogenic in origin, with most mortalities in care occurring as a direct result of the reason for admission, either by euthanasia or unassisted death [3]. The geographical location of a WRC and the species admitted both influence outcomes [3]. Overall, similar outcomes are seen in the WRC internationally. In New York State, the overall release rate for animals receiving care was 50.2%, while 45.3% died or were euthanised [33]. In New South Wales, Australia, 55% of animals survived, including 37.1% that were released, and 45% died [2]. In Queensland, Australia, ‘mortality’ was listed as the outcome for 57.4% of animals, with ‘positive outcomes’ in the remaining 42.6% of admissions [16]. In Catalonia, Spain, 63% of animals were released, 13% were euthanised, 22% died, and 1% were kept in captivity [52]. In Italy, 53.9% of animals were released, 4.7% were euthanised, and 41.4% died [87], and in Greece, 55.8% were released, 5.6% were euthanised, and 27.9% died [12].

In this study, significantly more birds were euthanised than mammals, and significantly more mammals were released than birds. These findings differ from previous data from across the UK, where 47% of birds and 31% of mammals were released [46]. The differences between this study and the previous UK study may have arisen because of changes in admissions and care over time, or there may be geographical impacts on both the species admitted and the reasons for admission, or the differences may result from WRC biases in the care provided. In a review of international WRC outcomes, mammals and birds were found to be equally likely to survive all stages of rehabilitation, although survival rates varied between locations [3]. In Ontario, Canada, reptiles had a higher rate of release (63.6%), compared with birds (48.3%) and mammals (42.1%) [99]. In New South Wales, reptiles again had the best outcomes (57.7% released), followed by birds (37.7% released) and mammals (28.4% released) [2]. In Chile, 40.7% of reptiles, 25.4% of mammals, and 19.5% of birds were released [36]. In New York State, amphibians had the highest release rate (57.3%), followed by mammals (54.5%), reptiles (47.5%), and birds (46.9%) [33]. In Queensland, however, the greatest number of positive outcomes were seen in mammals (marsupials at 50.1% and eutherian mammals at 58.1%), then birds (44.3%), and finally reptiles (42.6%) and amphibians (32.1%) [16]. In Gran Canaria, 54% of birds were released [53]. In Greece, reptiles, amphibians, and songbirds had the highest release rates, and mammals had the highest death rates [12]. As is the case when comparing UK centres, several factors are likely to impact the release rates of different classes of animals between centres internationally [3], and this would benefit from further investigation.

Species-specific ecology, biology, and behaviour impact on the reason for admission of individual animals to WRCs [16], which in turn affects the severity of injury or illness and the likelihood of release [43]. The wide variety of species around the world makes international comparisons more difficult, but some similar taxonomic trends are recognised [3]. The availability and quality of both veterinary and wildlife rehabilitator care are also likely to be significant factors. Further analysis of species-specific trends in this study would be beneficial, both to help improve standards of triage and care and to inform conservation strategies.

The high number of animals, across all classes, that require euthanasia in WRCs can be emotionally difficult for both veterinary professionals and wildlife rehabilitators to accept [5,6,7]. Early euthanasia is, however, necessary to minimise the negative welfare impacts of captivity and/or an unsuccessful release [3].

### 4.11. Reason for Admission vs. Outcome

The reason for admission had a significant impact on the eventual outcome for the casualty and the likelihood of release; ‘orphaned’ animals had a much better prognosis than ‘injured’ animals. These findings concur with those of other studies in which uninjured animals, in particular orphans and juveniles, across a range of species, had a better survival rate to release [2,16,17,52,61,62,99,100,101]. In contrast, admission to WRCs because of injury or disease generally has a much lower likelihood of survival [2,12,17,52,99]. Severity of injury is the most important factor affecting outcome [42,43], highlighting the need for a sound triage policy. In other studies, injury, specifically due to road traffic collisions, carried the poorest prognosis [16,27,95]. Attacks by dogs and cats were the second most common reason for mortality in several studies [16,95], with mortality rates of 69–78% quoted in several studies [10,16,86,87,89].

Young animals, in addition to healthy confiscated animals in some countries, carried the best prognosis for release. Euthanasia, or natural death, is, however, a frequent outcome for the many animals admitted to WRCs following trauma. WRCs can successfully care for animals, but ensuring that this is not at a cost to animal welfare is essential. Published data can help focus resources and manage the expectations of veterinary and WRC staff, encouraging prompt euthanasia where appropriate.

### 4.12. Outcomes over Time

In this study, the decision to euthanise an animal at SWWR was based upon careful examination and a ‘triage’ decision following veterinary advice. ‘Triage’ of wildlife casualties is the process of decision making in terms of whether to treat or euthanise [102] and should ideally take place within as short a time as possible following admission, ideally within 48 hrs [62,102]. Only animals with a good chance of survival back in the wild should be rehabilitated [43]. During the study period, protocols for examination, triage, and euthanasia at SWWR were increasingly formalised [103], alongside additional staff training. Staff retention over this time was also felt to be good, ensuring that more experienced staff were increasingly involved in decision-making processes. Although these trends were not analysed and there are no formal data to support these hypotheses, these changes are thought to have contributed to the significant increase in the proportion of animals euthanised and a similar significant reduction in the proportion of animals dying in captivity. Despite these changes, the proportion of animals released remained stable over time. This is very similar to the impacts of improved triage shown in a study of woodpigeons [62]. A systematic review of the literature [3] showed study location was a strong predictor of death in care because of differing triage and euthanasia protocols. Unassisted (‘natural’) deaths in care are an indicator of poor-quality triage criteria, treatment, and husbandry protocols [3]. The success of treatment and rehabilitation of wildlife casualties depends on the facilities, suitably trained personnel, veterinary services, adequate funding, and availability of release sites [44]. The availability of a good standard of veterinary care, with appropriate facilities and equipment, is very variable within the UK and likely to be a limiting factor in other lower-income and more geographically challenging regions. Veterinary care is, however, only part of the wildlife care process, and the provision of good ongoing care, together with experienced staff and appropriate rehabilitation facilities to ensure true fitness prior to release, is also required [66]. As well as care and facilities influencing wildlife triage decisions, other factors such as local culture (for example, around euthanasia), awareness and attitudes to animal welfare, and the relative value of the animal may also be influencing factors [30,44,104]. 

### 4.13. Time in Captivity

In this study of releases from a WRC, the mean duration of time in captivity was heavily skewed by a small number of dependent young animals, especially mammals, staying in the centre for a long period of time prior to release. The aggressive triage policy on admission also meant that many cases had an outcome reached on day 0, of which most were euthanised. Animals that were euthanised spent the shortest time in captivity, animals that died despite attempts to treat them spent a longer time in captivity, and animals that were eventually released spent the longest time in captivity. A comparable Australian study had no dissimilar findings; 64% of animals were in WRCs five days or less before an outcome, 8% were kept up to 10 days, 14% up to 25 days and 7% were still in captivity at 100 days [4]. 

The clear aim of the triage and rehabilitation process is to minimize the necessary time in captivity, especially for any animals that will not survive the process, whilst at the time ensuring those released have a good chance of survival back in the wild; this is a very careful balancing act. Concern has been expressing regarding lengthy periods in captivity and the impact of that on long-term survival [4]. Habituation to humans and loss of wild behaviours such as predator avoidance may result in poor survival. However, for young animals, time in captivity is needed for hunting, foraging, and learning wild behaviours [3]. Time in captivity can, however, benefit animals that require translocation prior to release [105]. There are some possible pitfalls for individual animal welfare associated with captivity, including stress [22,23,24], development of physical problems during rehabilitation (e.g., Pododermatitis; [106]), spread of disease within WRCs (e.g., Dermatophytosis; [25]), and inappropriate behaviour upon release [107]. The true cost–benefit of time in captivity, when compared to survival in naturally occurring non-rehabilitated populations, is probably unknown, as post-release monitoring in most centres is poor [3]. Other authors have also rightly questioned the economic cost–benefit of treating some species and types of injury for protracted periods of time [52] and this is a significant factor for all self-funding WRCs.

### 4.14. Importance of Findings and Opportunities for Further Study

The true success of wildlife rehabilitation is variably defined by the different stakeholders involved [6,8,9]. It may be judged in terms of preventing unnecessary suffering through euthanasia [108], the number of casualties rehabilitated and released [6], and by the number surviving in wild post release compared to ‘wild’ counterparts [3]. 

Poor records in many wildlife centres result in a lack of reliable information [3,4,44]. The quality of data may also vary depending upon the expertise of those making the records [42,43] and the transparency and honesty of the outcome figures released [6]. Standardisation of data collection methods (e.g., WILD-ONe in North America; [10]) would be of benefit in this respect. 

There is a clear benefit in the availability of basic data to allow for reflection on protocols, comparison with other WRCs, appropriate planning and allocation of resources for facilities and equipment, and facilitation of evidence-based practices. Compared to other areas of veterinary medicine, wildlife rehabilitation lacks strong scientific evidence on which to develop best practice guidelines, and this potentially has negative consequences for animal welfare. Support for those involved in wildlife rehabilitation, in the form of training and mentoring, is still lacking, as has been noted by others [2,5,7].

Understanding the long-term impacts of wildlife rehabilitation necessitates detailed post-release monitoring. Without knowing how long rehabilitated animals survive after release, compared to their wild counterparts, the whole process of wildlife rehabilitation, the financial costs associated with it, as well as the potential welfare harms to both people and animals can all be criticised. The current study, like many before it, fails to address these concerns as it is unable to provide data on the longer-term outcomes for released animals. Future studies should better consider the animal welfare impacts of captivity within a WRC, as well as survival post-release and the potential associated harm to welfare. Several factors have been shown to impact the long-term survival of rehabilitated wildlife, including species of animal, timing of release, release method, quality of the release habitat, and presence of predators [3]. New methods of animal welfare monitoring, for example, behavioural and physiological indicators of stress in captivity [24], alongside technological advances in pre- and post-release monitoring, should help to improve such investigations in the future.

There is increasing evidence that wildlife rehabilitation can have conservation population-level effects by supporting in situ recovery and enforcing declining populations [20,21,109]. Perhaps more importantly, from both an animal welfare and conservation perspective, are the indirect benefits of wildlife rehabilitation such as education to reduce human–wildlife conflicts [6,10,12,18,40,110]. WRC data can have a sentinel role for anthropogenic drivers, pressures and impacts such as environmental toxicities [14] and urbanisation [17]. It is important, however, to acknowledge that wildlife rehabilitation does not necessarily reflect the main natural causes for morbidity and mortality, but instead reflects areas of human–animal conflict [69]. 

As well as impacts upon individual animals, the possible adverse welfare consequences on other wild animals and the environment should be considered before animals are released. These may include human–wildlife pathogen transmission before release [29,111], the transmission of pathogens to other species after release [26], direct trauma to other wildlife from veterinary treatments including euthanasia drugs [112], and dissemination of antimicrobial resistance [28]. Conflict with humans by animals ‘in the wrong place’ [113] and following habituation in WRCs [107] may also lead to welfare harm. Further research is needed to better understand the impacts of WRC releases on wild animal welfare.

The mental health impacts of wildlife care should not be underestimated. Psychological and financial pressures associated with workload, working conditions, difficult decision-making and compassion fatigue have been described by other authors in both wildlife rehabilitators [2,5,6,7,44,114] and veterinary professionals [2,7,30,31] but were not considered in this study. Any future studies should be designed to include both the potential animal and human harms associated with wildlife rehabilitation, without which a balanced assessment of the benefits cannot be made. Sound evidence for wildlife rehabilitation practices, underpinned by more research where necessary, will guide such support, training, and best practices.

## 5. Conclusions

Wildlife rehabilitation can benefit animal welfare through appropriate treatment, rehabilitation, and release of either debilitated or orphaned animals. There is also, however, the possibility of welfare harms through inappropriate treatment of individuals and the stress of prolonged captivity. Triage, with early euthanasia where necessary, has been shown to be of critical importance both in this and in earlier studies [3,62,102]. 

Veterinary and wildlife rehabilitator education in the UK should focus on the species and conditions likely to be most frequently encountered, including neonatal care and treatment of injuries in birds and hedgehogs. For veterinary professionals, examination, triage, and trauma management are more important than an extensive knowledge of wild animal diseases. Knowledge of species-specific ecology and biology is, however, required to make triage decisions. For wildlife rehabilitators, skills in rearing, rehabilitating, and releasing young animals are most important. A clear seasonality to admissions is reported, and this can be used to plan staffing and resources.

Injury resulting from direct anthropogenic activities is a more common reason for admissions to WRCs than naturally occurring disease and carries a poor prognosis. Examination and primary triage must ensure only those animals likely to be released are treated, which often necessitates a protracted time in captivity. Professional veterinary involvement is essential to both assess and appropriately treat challenging trauma cases. To be successfully released back to the wild, animals will require intensive rehabilitation at a professionally staffed and well-resourced WRC. Euthanasia will often be the most welfare-positive outcome, and the expectations of both veterinary professionals and wildlife rehabilitators need to be managed in this respect, with appropriate psychological support. 

Public education and engagement is an essential part of reducing anthropogenic WRC admissions and, alongside using data to influence policy, could be more important than treating individual animals.

## Figures and Tables

**Figure 1 animals-14-00086-f001:**
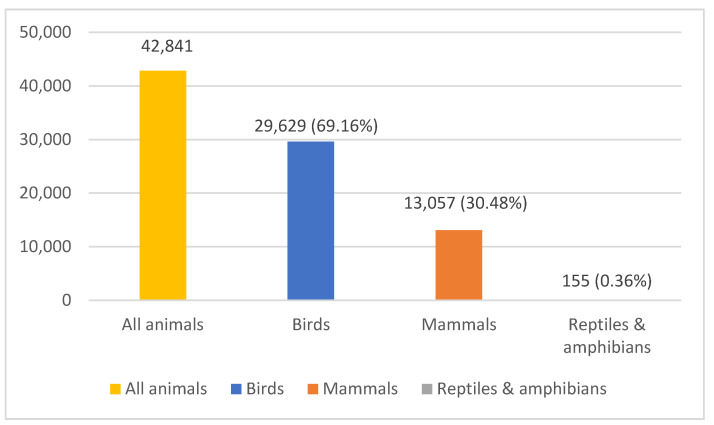
Bar chart illustrating the class of animals admitted over a ten-year period.

**Figure 2 animals-14-00086-f002:**
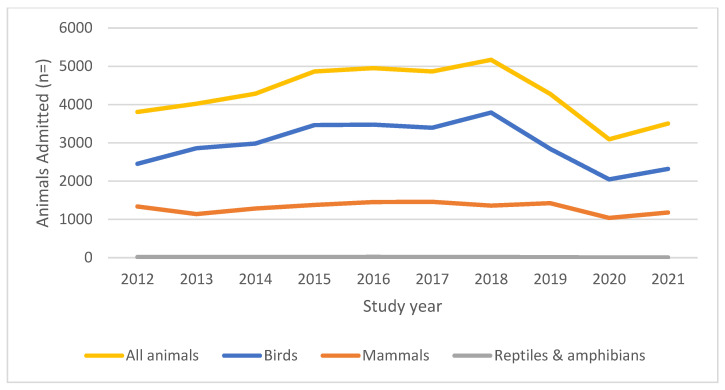
Graph illustrating the number and class of animals admitted by study year.

**Figure 3 animals-14-00086-f003:**
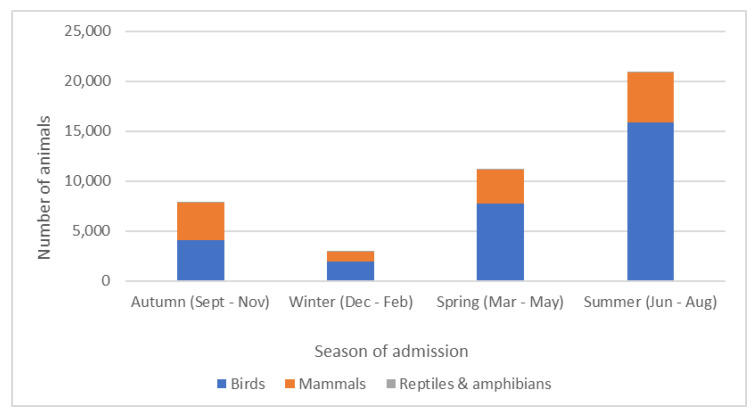
Stacked bar chart illustrating the seasonality of admissions for all animals and the proportion of animals in each class.

**Figure 4 animals-14-00086-f004:**
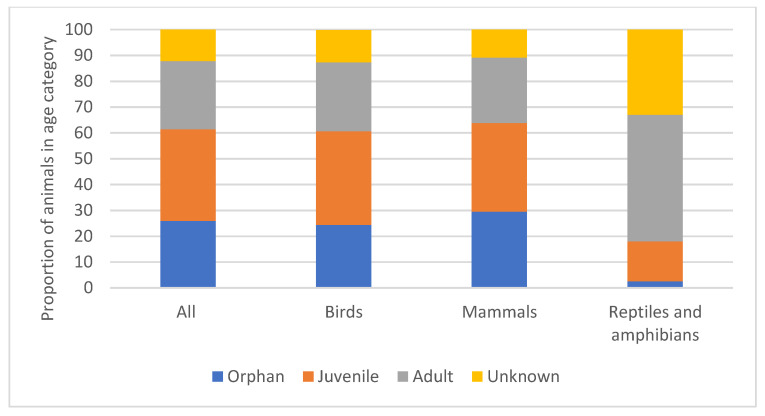
Stacked bar chart illustrating the proportion of animals admitted in each age class.

**Figure 5 animals-14-00086-f005:**
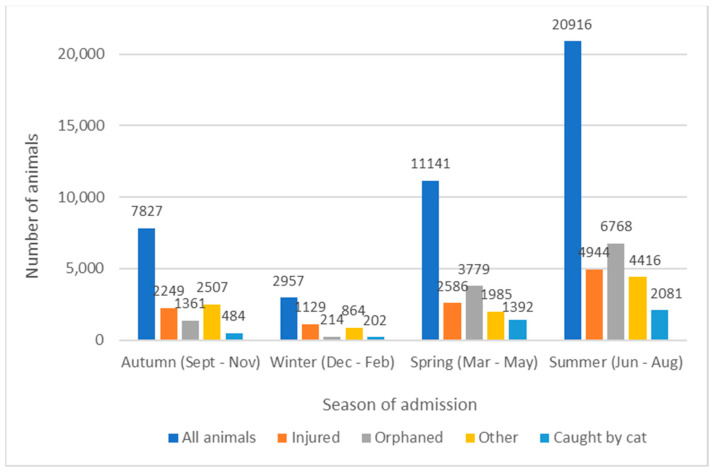
Bar chart illustrating the seasonality of admissions for all animals and the four most common reasons for admission.

**Figure 6 animals-14-00086-f006:**
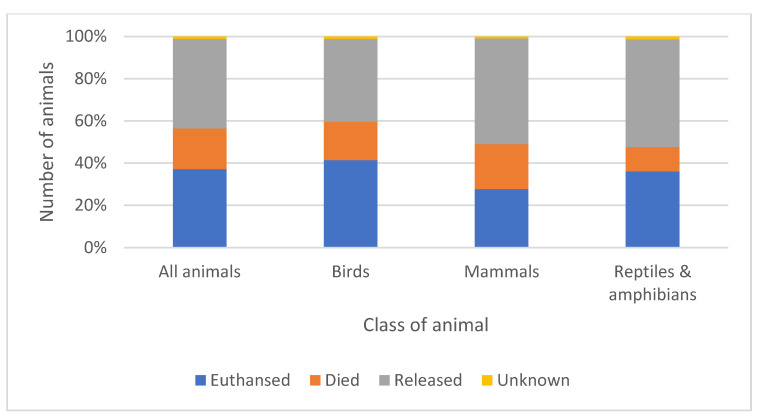
Stacked bar chart illustrating the number of animals in each outcome category for all animals and each animal class.

**Figure 7 animals-14-00086-f007:**
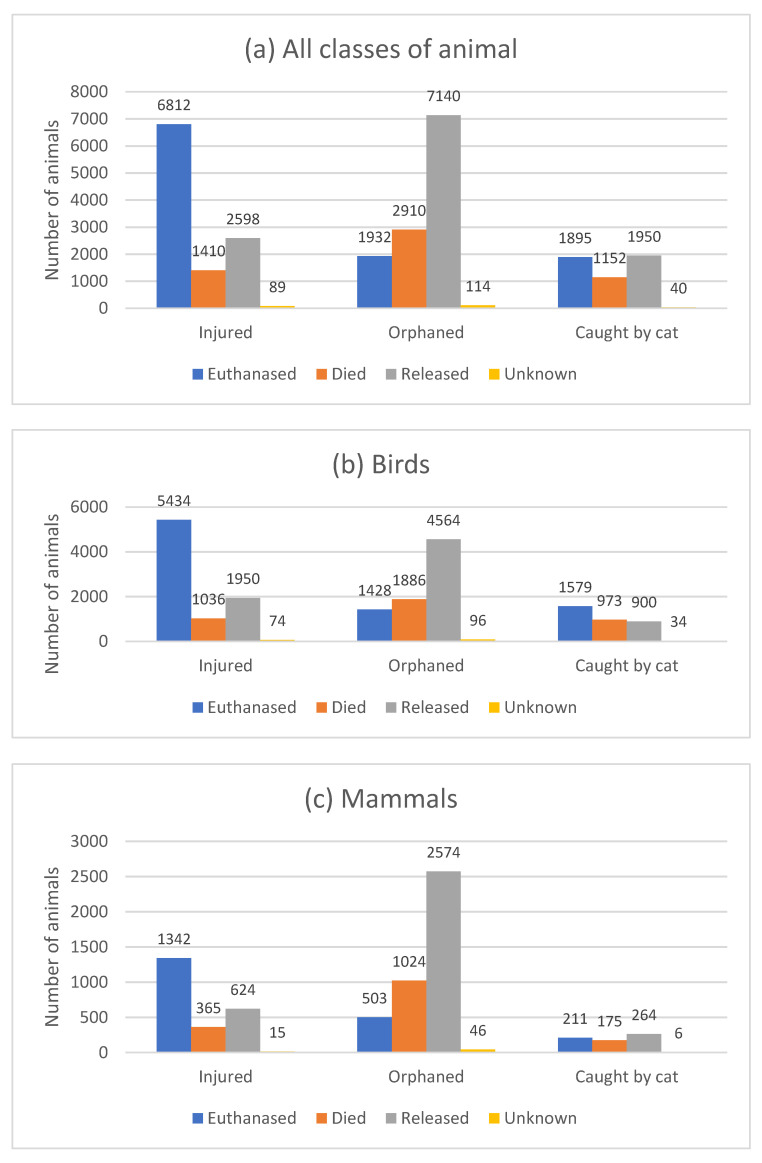
Bar charts illustrating the final outcome for the top three known reasons for admission—‘injured’, ‘orphaned’, and ‘caught by cat’: (**a**) for all classes of animal, (**b**) birds, (**c**) mammals, and (**d**) reptiles and amphibians.

**Figure 8 animals-14-00086-f008:**
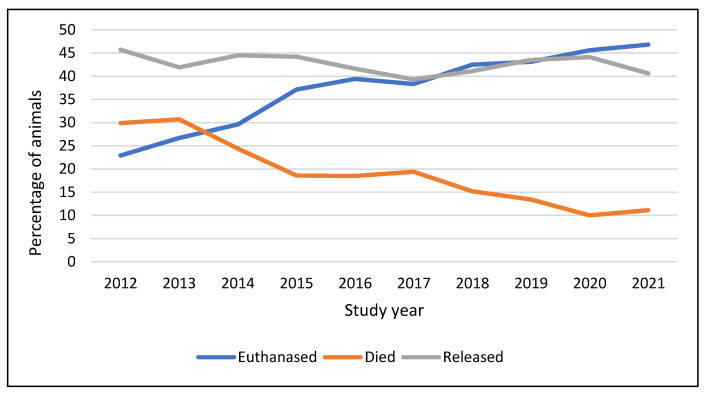
Graph illustrating the percentage of animals euthanised, dying naturally, and released over the 10-year study period.

**Table 1 animals-14-00086-t001:** Year-on-year percentage change of total birds and mammals admitted.

Year-to-Year Transition	% Change on Previous Year	
	Birds	Mammals
2012–2013	17%	−14%
2013–2014	4%	13%
2014–2015	16%	7%
2015–2016	–	6%
2016–2017	−2%	–
2017–2018	12%	−7%
2018–2019	−23%	4%
2019–2020	−28%	−27%
2020–2021	13%	13%

**Table 2 animals-14-00086-t002:** Year-on-year percentage change in the proportion of birds and mammals admitted and the Z statistic relating to the significance of the proportional change.

Year-to-Year Transition	Birds		Mammals	
	Change	Z Statistic	Change	Z Statistic
2012–2013	+7%	6.58 *	–7%	6.38 *
2013–2014	−1%	1.58	+2%	1.66
2014–2015	+1%	1.80	–2%	1.77
2015–2016	–1%	1.21	+1%	1.16
2016–2017	-	0.54	+1%	0.06
2017–2018	+3%	4.07 *	–4%	4.04 *
2018–2019	–7%	7.29 *	+7%	6.90 *
2019–2020	-	0.27	+1%	0.31
2020–2021	-	0.02	-	0.03

* *p* < 0.00001.

**Table 3 animals-14-00086-t003:** Season-to-season percentage change in the proportion of birds admitted and Z statistic relating to the significance of proportional change.

Season-to-Season Transition		
	Change	Z Statistic
Autumn to Winter	+13%	12.30 *
Winter to Spring	+5%	5.25 *
Spring to Summer	+6%	12.36 *
Summer to Autumn	−24%	40.30 *

* *p* < 0.00001.

**Table 4 animals-14-00086-t004:** Species (common and Latin names) with more than 1000 identified individuals admitted over the 10-year study period.

Common Name	Latin Name	UK Conservation Status	Number
Hedgehog-European	*Erinaceus europaeus*	Mammal Society Red List ^1^	5972
Pigeon-Wood	*Columba palumbus*	BTO Amber List ^2^	3737
Gull-Herring	*Larus atgentatus*	BTO Red List ^2^	3659
Pigeon-Feral/Domestic/Racing	*Columba livia domestica*	BTO Green List ^2^	3115
Blackbird-Common	*Turdus merula*	BTO Green List ^2^	2340
Sparrow-House	*Passer domesticus*	BTO Red List ^2^	1703
Duck Mallard	*Anas platyrhynchos*	BTO Amber List ^2^	1436
Dove-Collard	*Streptoppelia decaocto*	BTO Green List ^2^	1334
Rabbit-European	*Oryctolagus cuniculus*	Mammal Society Green List ^1^	1303

^1^ [47] ^2^ [48].

**Table 5 animals-14-00086-t005:** Number and percentage of animals admitted according to reason for admission for each class.

	All Classes	Birds	Mammals	Reptiles/Amphibians
Injured	10,909 (25.5%)	8484 (28.7%)	2346 (18.0%)	69 (44.5%)
Poisoned/polluted	89 (0.2%)	54 (0.2%)	35 (0.3%)	0
Orphaned	12,124 (28.3%)	7974 (26.9%)	4147 (31.8%)	3 (1.9%)
Natural causes	1131 (2.6%)	619 (2.1%)	507 (3.9%)	5 (3.2%)
Other	9772 (22.8%)	5841 (19.7%)	3887 (29.8%)	44 (28.4%)
Caught by cat	4159 (9.7%)	3486 (11.8%)	656 (5.0%)	17 (11.0%)
Caught by dog	429 (1.0%)	184 (0.6%)	242 (1.9%)	3 (1.9%)
Unknown	4228 (9.9%)	2977 (10.0%)	1237 (9.5%)	14 (9.0%)

## Data Availability

The data presented in this study are available on request from the corresponding author. The data is not publicly available due to belonging to the WRC.

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
