# Peer review of "Trends in Admissions and Outcomes at a British Wildlife Rehabilitation Centre over a Ten-Year Period (2012–2022)"

_animals, 2023, doi:10.3390/ani14010086_

Round 1

Reviewer 1 Report

Comments and Suggestions for Authors

This is an interesting overview of the field of UK rehabilitation and the data and analysis used here are very useful to provide a summary of the recent trends.

I have highlighted some aspect that could improve the presentation and analysis of the data to give an more indepth and clearer picture.

Methods should include a data clean up/manipulation section which explains how you grouped and sorted this large data set.

There should be more detail about what was analysed including the use of chi squared tests. Why just focus on 2014-2018 for comparison of admissions in chi square.

Line 142- How were amphibians and reptiles distinguished if grouped together. Did you have to analyse species rather than class if so please explain this in the methods. Though only 9 herps could be identified to species level.

Please change pie charts to bar charts which are much better at displaying the differences?

Stacked bar charts are better for presenting proportional data when the totals of each group add up to 100. A line graph, showing the trends over time would be a better way to present the information in fig 2.

Figure 4 would be better represented as proportion of age class from each taxa group and would allow reptile and amphibians to be visible on the graph.

Line 174 please explain the term domestic/zoo as these are quite different categories.

Line 219 did any animals that were brought in, fall under the Invasive Species Act.

Chi squared test on line 223 - did you test this by proportions, given that there were more birds than mammals?

Line 256- not normally distributed - rather than non parametric which refers to a type of test. What test was used to ascertain this?- please add to the methods section. It would have been good to have seen how this data compares were there significant differences in time to final outcome, depending on injury type and outcome type and taxa group. Can you test this statistically?

Line 273- why would this centre be suitable to generalise back to wider UK rehabilitation centres?

Line 323- and also a factor of lower public outdoor pursuits during this type.- It would be interesting to note if admissions increased over the holiday period during the winter in comparison to other winter times.

The discussion is substantially long, especially in comparison to the introduction. Some information here could have been provided by way of introducing the topic - especially the different reasons for injury. The discussion could consider some of these factors together to provide a more holistic overview of UK wildlife rehabilitation.

Well written in general though some minor punctuation errors eg. around however 

Author Response

File uploaded

Reviewer 2 Report

Comments and Suggestions for Authors

It was a pleasure to review this manuscript.  I found it both interesting and informative.  I do, however, have a number of concerns.  

Let me begin with what I like about this paper.

It is exceptionally well-written and organized.  In over thirty years of peer-reviewing papers, I have rarely encountered one with this level of writing.  For this reason alone, these authors are to be praised. 

The Introduction and Discussion sections comprise a truly excellent review of the current state of Wildlife Rehabilitation Centers. 

They have provided, as was their goal, a thorough statistical description of the types of animals that the last decade has presented at their particular WRC in Somerset UK. 

Those positive observations notwithstanding, I have the following concerns. 

These authors have not made a strong case for why a reader will want to learn about the statistics in Somerset particularly.  Indeed, by the authors’ own excellent review, it appears that the pattern of admissions at Somerset pretty much matches what has been seen elsewhere.  I would not characterize this as a fatal flaw.  The authors have done an excellent job at describing their findings.  I am just asking that they make a stronger case for why their findings will be of interest to a general reader. 

Much of the Discussion is repetitive of what has already been presented in the Results.  The overall format of each Discussion section is to repeat the principal findings, and then to compare those finding with what has been reported elsewhere.  The effect is that one is left re-reading most of the principal findings.  If the journal would allow, I think this paper would be improved by reorganizing the last two sections into a combined Results/Discussion section. 

In the Discussion, the authors miss opportunities to speculate about the significance of their findings.  For example, in Lines 288-303 they point out that patterns vary by location.  But why?  These authors can/should offer their own views about why this might be so.  But this is just one example.  Their own views can/should be offered each time their results are compared to what has been reported elsewhere. 

Lines 311-312 discuss the covid/post-covid time periods.  I would encourage these authors to discuss this more thoroughly.  If, as they say, more people spent time in the countryside, why/how did this effect the visibility/discovery of wildlife cases in need? 

Lines 735-778 provide a very good discussion of three very important measures:  survival rates after release, animal welfare during rehabilitation, and possible psychological effects on caregivers.  However, this study does not address any of them.  This should be taken as an opportunity to (a) admit that as absent in the present report, and (b) provide a recommendation for such variables to be investigated in the future.  Indeed, these authors exhibit such acumen for this discipline, it would be gratifying to see them report that studies of that type will be forthcoming from their own institution.  

The Chi Square procedure is appropriate for comparing head-count data of the type presented here.  However, I do want to make two points about the statistical tests presented.  One, it is not ordinary or acceptable to only present selected Chi Square findings.  That is, it is expected that the results of every comparison be provided (in a Table usually).  Nearly all of them, I suspect, will be non-significant.  That is not something that should be buried.  Two, the Chi Square is less useful for multi-factorial comparisons, such as assessing the statistical significance of Animal-Class by Season.  The use of a Generalized Linear Mixed Model (available in SPSS, for example) would be a better choice for that. 

Lines 305-306 refer to a general increase at the beginning of the decade, but this was not established as significance in Results. 

Lines 314-315 report that birds were not admitted in some years due to concerns about H5N1.  Since this effected the very counts reported here, this fact should be in the Methods section. 

Lines 679-704 point out that an important trend over years was due to changes in facility practices.  This too should be in the Methods section.  The trend over time was created by their own procedure.  It should not be reported as a finding! 

In an effort to be helpful, I also offer the following:

Line 116 “trained” should be defined/explained.

Line 118 “unsuitable” should be defined/explained. 

Line 134 “long-term captivity” should be defined/explained. 

Line 228 I believe that “know” should be “known”

Line 254 I think that “was” might better be “ranged”

Line 398 I believe “can” might better be “could”

Line 783 makes an important claim about the importance of triage.  However, no reference is provided to support this claim. 

Lines 141, 196, 216, 482 start sentences with numerals.  This, I believe, is not permitted in most English style guidelines.  That is, it is acceptable to write ‘Cases amounted to 42 instances of intensive care’ or to write ‘Forty-two intensive-care cases were counted.’  But it is not permitted to write ’42 intensive-care cases were counted.’

Having said all of that, I would hope that these authors not be discouraged.  Indeed, this manuscript already has the makings of a truly excellent report.  I hope that the issues raised here can be addressed so that it can eventually be published.  It is already of such high quality, it can/should be available to a general readership. 

Author Response

Uploaded

Round 2

Reviewer 2 Report

Comments and Suggestions for Authors

In their revision, the authors have addressed many of my concerns.